# Intelligent Crack Detection Method Based on GM-ResNet

**DOI:** 10.3390/s23208369

**Published:** 2023-10-10

**Authors:** Xinran Li, Xiangyang Xu, Xuhui He, Xiaojun Wei, Hao Yang

**Affiliations:** 1School of Rail Transportation, Soochow University, Suzhou 215006, China; 20234046002@stu.suda.edu.cn; 2School of Civil Engineering, Central South University, Changsha 410075, China; xuhuihe@csu.edu.cn (X.H.); xiaojun.wei@csu.edu.cn (X.W.); 3School of Transportation and Civil Engineering, Nantong University, Nantong 226019, China; h.yang@ntu.edu.cn

**Keywords:** road crack detection, ResNet, GAM, focal loss, leaky ReLU

## Abstract

Ensuring road safety, structural stability and durability is of paramount importance, and detecting road cracks plays a critical role in achieving these goals. We propose a GM-ResNet-based method to enhance the precision and efficacy of crack detection. Leveraging ResNet-34 as the foundational network for crack image feature extraction, we consider the challenge of insufficient global and local information assimilation within the model. To overcome this, we incorporate the global attention mechanism into the architecture, facilitating comprehensive feature extraction across the channel and the spatial width and height dimensions. This dynamic interaction across these dimensions optimizes feature representation and generalization, resulting in a more precise crack detection outcome. Recognizing the limitations of ResNet-34 in managing intricate data relationships, we replace its fully connected layer with a multilayer fully connected neural network. We fashion a deep network structure by integrating multiple linear, batch normalization and activation function layers. This construction amplifies feature expression, stabilizes training convergence and elevates the performance of the model in complex detection tasks. Moreover, tackling class imbalance is imperative in road crack detection. Introducing the focal loss function as the training loss addresses this challenge head-on, effectively mitigating the adverse impact of class imbalance on model performance. The experimental outcomes on a publicly available crack dataset emphasize the advantages of the GM-ResNet in crack detection accuracy compared to other methods. It is worth noting that the proposed method has better evaluation indicators in the detection results compared with alternative methodologies, highlighting its effectiveness. This validates the potency of our method in achieving optimal crack detection outcomes.

## 1. Introduction

The escalating demands on transportation and the pervasive impact of the natural elements, such weather conditions, has made the occurrence of cracks within transportation infrastructure (such as bridges, tunnels and highways) a prevalent concern. If the challenge of crack detection is not proactively addressed, it could precipitate a diminution in the service life of the infrastructure. Even more concerning, it might trigger a sequence of safety-related incidents, exacting substantial economic ramifications and human casualties [1]. This underscores the imperative for diligent road inspection and maintenance, with crack detection forming a pivotal facet of the broader road maintenance framework [2]. The prompt identification and subsequent remediation of cracks via specialized detection methods bear profound practical significance. This proactive approach not only mitigates road safety risks but also yields tangible cost savings within road maintenance endeavors [3].

The conventional approach to road crack detection relies primarily on manual inspection, where personnel visually examine road cracks. However, this method proves time-consuming and labor-intensive, often yielding unsatisfactory outcomes due to environmental challenges, such as dim lighting, shadows and background noise [4]. Contemporary road crack detection techniques can be divided into two types: image processing techniques and deep-learning-based techniques. Image processing techniques encompass methods based on the grayscale threshold [5,6] and edge detection [7,8]. Nonetheless, their effectiveness diminishes when confronted with intricate crack images characterized by rich details, leading to diminished detection precision. Amid the rapid evolution of computer vision, deep learning approaches have gained prominence in domains spanning road crack detection, medical image processing, natural language processing and object detection due to their excellent automatic feature extraction capabilities and feature representation generalization capabilities and have achieved a series of remarkable results [9,10,11,12]. Foremost among these methodologies is the convolutional neural network (CNN). In the domain of road crack detection, the CNN autonomously extracts profound features from crack image datasets, employing its intricate network architecture to classify cracks and backgrounds, thus delivering more dependable outcomes in crack detection. Zhang et al. [13], for instance, achieved remarkable detection accuracy by training a deep CNN on collected crack images. Cha et al. [14] designed a vision-based technique, mainly through the deep architecture of a CNN combined with a sliding window method to achieve concrete crack detection. Chen et al. [15] developed a detection framework that melds a CNN with Naive Bayesian decision making for video frame analysis and achieved good results in nuclear power plant reactor crack detection. Liang et al. [16] designed a dual CNN composed of a CNN and full convolutional network to automatically and efficiently detect the presence of cracks in concrete bridges, achieving high detection accuracy with minimal noise interference.

While CNNs proficiently extract crack image features and achieve precise road crack detection, its constrained representation capacity poses challenges in maintaining generalization across varying object scales, sizes and angles. Addressing this, the attention mechanism can allow the model to dynamically modify the weight based totally on the importance of the input data so that it can focus on processing some critical feature information that helps with the detection task, reducing attention on irrelevant redundant feature information, andthus, enhancing the generalization fitting ability and detection performance [17,18]. The attention mechanism has been efficiently utilized in different fields, such as object detection [19], image processing [20], fault diagnosis [21] and electrocardiogram detection [22]. Regarding crack image data, an imbalanced distribution between cracks and backgrounds often occurs, and the traditional detection algorithm usually performs follow-up detection work under the premise of category balance. However, if the category is not balanced, it will make it more difficult for the model to distinguish cracks and backgrounds, resulting in poor model detection performance and unsatisfactory detection results. In order to tackle this, we put forward a novel road crack detection network framework based on a GM-ResNet, which is constructed with ResNet-34 as the basic network framework. By introducing the global attention mechanism (GAM) attention module [23], the model can capture the global and local feature information in crack images better; by relying on the channel and the spatial attention mechanisms, the model pays attention to more important feature channels, the importance of the detection target in the entire image is obtained and feature discriminability and detection efficacy are boosted. Moreover, rectifying class imbalance in crack images is a pivotal question. Focal loss [24] is introduced as the training loss function of the GM-ResNet, and the weight of the loss function and the attention to positive and negative samples, as well as difficult-to-classify samples, are adjusted to deal with the negative influence of class imbalance on detection performance by controlling the two hyperparameters in focal loss. Finally, by substituting the fully connected layer with a connected neural network (MFCNN), the deep network constructed by multilayer linear, batch normalization and activation function layers can respond well to this trouble of poor model performance in complex image data. Furthermore, the leaky rectified linear unit (ReLU) is used as the activation the function of the GM-ResNet to avoid the zero gradient challenges associated with the traditional ReLU function and bolster the nonlinear fitting performance of the proposed model. The example verification of the crack image datasets shows that the model surpasses other comparative models in terms of detection accuracy and visualization effect, which affirms the effectiveness and superiority of the GM-ResNet model in the domain of road crack detection.

The structure of this paper is as follows: in the second section, the relevant knowledge and the proposed road crack detection architecture of the GM-ResNet are described in detail, and the third section undertakes experimental validation, presenting results from the proposed model and analogous comparative models. The fourth section summarizes the entire paper and prospects for follow-up work.

## 2. The Proposed GM-ResNet Road Crack Detection Framework

This section starts with ResNet and introduces the concepts of the GAM and the proposition of a MFCNN as a replacement for the ResNet fully connected layer. Subsequently, the focal loss function is introduced. Finally, by amalgamating the GAM and MFCNN within the ResNet-34 framework, we present a novel GM-ResNet architecture tailored specifically for the task of road crack detection.

### 2.1. ResNet

The rapid evolution of deep learning technology has propelled the widespread utilization of a CNN as an exemplary network model. Its pervasive influence spans across a spectrum of domains, including facial recognition [25], intelligent transportation systems [26] and cancer classification [27], and the fields of structural health monitoring [28] and fault diagnosis [29]. The ubiquity of CNNs in these realms substantiates their pivotal standing as an architectural cornerstone within the deep learning domain. Emerging as an unparalleled and triumphant paradigm, CNNs have assumed the mantle of the most extensively adopted and efficacious deep learning network model. The breakthrough by AlexNet [30] in 2012, marked by extraordinary performance on expansive image datasets, ignited a transformative wave in the trajectory of deep learning. This pioneering success catalyzed the formulation of a suite of innovative and exceptional variants of CNNs, such as GoogleNet [31], VGGNet [32], DenseNet [33] and SqueezeNet [34], each making indelible marks across diverse tasks, and emerging as emblematic exemplars, showcasing remarkable accomplishments in distinct arenas. Many experts and scholars are committed to improving its feature representation ability by continuously stacking deeper network architectures to achieve better performance for the model. However, as the depth of the model increased, a perplexing observation emerged: contrary to expectations, the model performance exhibited diminishment. The reason for this was that gradient vanishing and explosion within the intricate network of hidden layers and the troubling issue of network degradation collectively contributed to the pronounced deterioration of the model’s efficacy. The proposal of ResNet [35] successfully solved this challenge. The ResNet, notably diverging from a conventional CNN, adds the output of all the layers before the active function to the output of the current layer through the skip connection structure of the residual blocks. This strategic fusion empowers the network to transmit the gradient directly to the previous network through the skip connection during backpropagation, effectively mitigating the issue of network degradation and thereby avoiding the problems of gradient explosion and disappearance during the training process of the model, which resulted in poor model performance.

Assuming that X denotes the original input data of the model and $F\left(X\right)$ is the output after passing through the residual block, the calculation formula of the identity map, $H\left(X\right)$, is as follows:(1)$$H\left(X\right)=F\left(X\right)+X$$

By using the equation above, an identity map can be constructed to convert the function, $H\left(X\right)$, required for the training and fitting of the original model into $F\left(X\right)+X$. This simple addition can significantly enhance the model training speed and efficacy and precludes network degradation due to excessive layering. The residual block encompasses two distinct types: the basic block and the bottleneck. Figure 1 is the network structure diagram of the residual block.

### 2.2. GAM

The core concept of the attention mechanism resides in its ability to selectively emphasize relevant feature information while disregarding unessential inputs, thus pinpointing the pivotal feature representations within deep neural networks. This strategy effectively eliminates the detrimental impact of extraneous data on model training outcomes during the training process [36]. Some scholars have progressively integrated the attention mechanism into diverse domains and achieved superior performance over previous approaches. The unveiling of the transformer [37] in 2017 ushered in a watershed moment and propelled the innovative deployment of the attention mechanism. Subsequently, an array of novel and efficient attention mechanisms emerged, including Squeeze-and-Excitation Networks (SENet) [38], the Convolutional Block Attention Module (CBAM) [39], the Efficient Channel Attention network (ECA) [40] and the GAM. The SENet mainly considers the influence of the relationship between different feature channels on the feature extraction ability of the model and deriving the relevance of each feature channel through learned attention weights, which can increase attention to useful feature information and suppress feature information that is not relevant to the target task of the model. This is accomplished through the squeeze and the excitation modules, meticulously realizing the function intended. The CBAM shares conceptual commonalities with SENet, centering on curbing irrelevant and superfluous feature interferences across channels by gauging the significance of each channel. Moreover, the CBAM embraces the significance of spatial information, achieving this through a comparable learning technique. Similar methodologies are employed to extract feature space importance and suppress irrelevant data, thereby prioritizing salient information. Both SENet and the CBAM achieve better performance by developing more complex attention modules, but this will certainly make the model more complex and greatly increase the computational burden. In order to strike an optimal balance between performance and complexity, Wang et al. developed the ECA network, an ultra-lightweight attention module, which effectively improves the performance of deep CNNs. Moreover, due to the limited parameters of the ECA module, it does not impose a significant burden on the computational power required for model training. The ECA module also understands the necessity of preserving dimensional interactions and avoiding excessive dimensionality reduction by analyzing the structure of SENet, thus, a one-dimensional convolution operation with an adaptive convolutional kernel size is utilized to realize the information interaction between local channels. However, the attention mechanism modules mentioned above all have a common problem, which is that they fail to recognize the significance of preserving channel and spatial information, resulting in the loss of cross-dimensional information and the interaction between global spatial dimensions. Therefore, considering the necessity of cross-channel dimensional interaction, Liu et al. designed an efficient attention mechanism that can ingeniously capture valuable feature information in the channel, spatial height and spatial width dimensions concurrently, robustly enhancing the information exchange across the dimensions, called the global attention mechanism (here: GAM).

The GAM is refined and enhanced, building upon the foundational concepts of the CBAM, which can not only decrease the reduction and dispersion of the feature information, but also simultaneously accentuate the amplification of global dimensional interaction features [23]. For this reason, the GAM redesigns a new channel–spatial attention sub-module using a sequential channel spatial attention mechanism. The architecture of the GAM is thoughtfully illustrated in Figure 2.

Let the input feature map be $F_{input}\in R^{C\times H\times W}$, the middle output feature be $F_{middle}$ and the final output feature be defined as $F_{output}$, and then the processing process of the GAM is as follows:(2)$$\begin{array}{l} F_{middle}=M_{c}\left(F_{input}\right)⊗F_{input}\\ F_{output}=M_{s}\left(F_{middle}\right)⊗F_{middle} \end{array}$$
where $M_{c}$ and $M_{s}$ represent the channel and spatial attention map, respectively, and ⊗ represents the multiplication operation between feature elements.

The GAM manifests through two independent sub-modules: the Channel Attention Module (CAM) and Spatial Attention Module (SAM). The CAM can extract global feature information by preserving the information of three dimensions. It first performs dimension conversion, and then inputs the dimension-converted feature map into a multilayer perceptron to enhance the cross-dimensional channel–spatial information dependence, where the reduction rate of multilayer perceptron is r. Finally, the dimension is reversed and the output, $M_{c}\left(F_{input}\right)$, is activated by the leaky ReLU activation function. Figure 3 is a network structure diagram of the CAM. The exact computational formula is represented as Formula (3).
(3)$$M_{c}\left(F_{input}\right)=\mathrm{LeakeyReLU}\left(\mathrm{Rp}\left(\mathrm{MLP}\left(p\left(F_{input}\right)\right)\right)\right)$$
where p represents the dimension transformation and Rp represents the inverse dimension transformation.

In SAM, two 7 × 7 convolutional layers are used successively to reduce and increase the number of channels to achieve the purpose of centralizing spatial feature information. This strategic operation can reduce the amount of calculation and fuse spatial information. Meantime, the conventional maximum pooling layer is removed from the SAM to further preserve the spatially useful information feature map. Finally, the output, $M_{s}\left(F_{middle}\right)$, is obtained through leaky ReLU activation. The structure visualization of the SAM network architecture is shown in Figure 4, and the specific operations are as follows:(4)$$M_{s}\left(F_{middle}\right)=\mathrm{LeakeyReLU}\left(\mathrm{Conv}\left(\mathrm{Conv}\left(F_{middle}\right)\right)\right)$$

In the formula, Conv denotes the convolution operation with a convolution kernel size of seven.

### 2.3. MFCNN

Recognizing the fact that the fully connected layer in the ResNet model often fails to achieve satisfactory results in processing complex image tasks, we have innovatively substituted this layer with an MFCNN composed of multiple linear, batch normalization and activation function layers. The nonlinear expression ability of the model has been greatly improved by embracing a more profound network architecture and harnessing the potency of the leaky ReLU nonlinearity. Consequently, it can fit complex data distributions and patterns better. Furthermore, the infusion of multiple intermediate layers imparts the model with the capacity to glean multifaceted beneficial features across various levels and strengthen its feature extraction and representation capabilities. The batch normalization layer also helps to prevent overfitting during model training, increasing the crack detection effect and generalization ability. The specific calculation rules for an MFCNN are as follows:(5)$$\begin{array}{l} F_{1}=\mathrm{LeakyReLU}\left(\mathrm{BN}\left(\mathrm{Linear}\left(F^{\prime }\right)\right)\right)\\ F_{2}=\mathrm{Linear}\left(\mathrm{LeakyReLU}\left(\mathrm{BN}\left(\mathrm{Linear}\left(F_{1}\right)\right)\right)\right)\\ \mathrm{LeakyReLU}\left(x\right)=\left\{\begin{array}{l} x,\;x>0\\ \alpha x,\;x<0 \end{array}\right. \end{array}$$
where $F_{1}$ is the middle transition feature, $F_{2}$ is the output feature processed by an MFCNN, BN represents the batch normalization operation and Linear represents the linear layer operation. Figure 5 shows the MFCNN network structure diagram.

### 2.4. Focal Loss Function

It is widely acknowledged that within a dataset containing crack images, the number of normal images (negative samples without cracks) will be much more than that of abnormal images (positive samples with cracks), and the proportion of cracks in the same image is often very small. At this time, the negative sample portion of the image without cracks and the image without cracks will become the majority category, while the positive sample portion of the image with cracks and the image with cracks will be the minority category, which leads to the occurrence of class imbalance. The traditional cross-entropy loss function has proven its efficacy in image processing tasks. However, in situations characterized by such class imbalances, employing the standard cross-entropy loss may inadvertently skew model predictions towards the majority categories while neglecting the minority categories, which may limit the performance of the model and result in unsatisfactory detection performance. Therefore, we use the focal loss function for model training to alleviate the adverse effects of category imbalance. The focal loss function incorporates adjustable weighting factors, which enable the model to effectively adjust the weight of easily classified samples (non-cracked images) during the training process and reduce its contribution to the loss by reducing the weight. Conversely, increasing the weight of difficult-to-classify image samples (images with cracks) makes the model pay more attention to such image samples, which helps to enhance the generalization learning ability of the model for difficult-to-classify image samples. The following is the calculation rule of the focal loss function:(6)$$\begin{array}{l} Loss_{FL}=-\alpha_{t}{\left(1-p_{t}\right)}^{\gamma }\log\left(p_{t}\right)\\ p_{t}=\left\{\begin{array}{ll} p & \;\;\;\mathrm{if}\;y=1\\ 1-p & \;\;\;\mathrm{otherwise} \end{array}\right. \end{array}$$
where $\alpha_{t}$ represents the weight factor of cross-entropy, which is adopted to balance the problem of an uneven proportion of positive and negative samples. The more negative samples, the smaller $\alpha_{t}$. γ denotes the adjustment factor applied to balance the samples for difficult or easy classification. γ>0 can reduce the loss of easily classified image samples and make the model pay more attention to image samples that are difficult to classify. And the values of *y* are 1 and −1, representing the foreground and background, respectively.

In the final analysis, the focal loss function can help the model better handle the troublesome question of uneven positive and negative image samples and imbalanced image samples that are difficult or easy to classify by introducing two regulatory factors, $\alpha_{t}$ and γ, when dealing with category imbalance, enhancing the classification accuracy of the cracked image samples and thereby enhancing the general performance of the model.

### 2.5. GM-ResNet Network Framework for Road Crack Detection

Based on the work above, in order to better realize crack detection and boost the detection performance of the model, we selected ResNet-34 as the backbone network and introduced the GAM into the ResNet-34 model. This pivotal inclusion enables comprehensive global cross-dimensional interactions across both the channel and spatial dimensions and ensures that the model can capture prominent three-dimensional features of the channel, such as its spatial height and width. This synergistic augmentation bolsters the model’s feature extraction potency significantly, thereby elevating the overall crack detection efficacy. Subsequently, we engineered an enhancement by substituting the conventional fully connected layer in ResNet-34 with the meticulously described MFCNN module outlined in Section 2.3. This augmentation empowers the model with a more profound capacity to accommodate image data nuances through more intricate network architecture, bolstered by the integration of batch normalization operations and activation functions. This further enhances the model’s feature extraction expression ability and reliability, effectively avoiding the occurrence of an overfitting phenomenon in the training process of the model, and improves the detection accuracy of the model. Finally, the focal loss function is used as the target loss function for model training. By dealing with the imbalance between positive and negative image samples and difficult- or easy-to-classify image samples, one can increase the weight of image samples that are difficult to classify and negative image samples. Thus, the model can distinguish them well, avoiding problems such as the decreased training accuracy and unsatisfactory detection performance caused by a class imbalance in model training. In addition, the nonlinear activation function leaky ReLU is used as the training loss function of the model, which avoids the occurrence of neuron death during training and increases the nonlinear feature expression ability of the model. Figure 6 delineates the architecture of the GM-ResNet, providing a comprehensive visualization of our approach. Transitioning to the actual application background of road crack detection, a flow diagram of the process of the road crack detection model based on the GM-ResNet is depicted in Figure 7, and its specific implementation steps are as follows:(1)The crack image dataset containing cracks with images and without images is divided into training and testing datasets according to a certain ratio.(2)The training datasets are input into the proposed GM-ResNet for model training, and the loss function of the training process adopts the focal loss function.(3)If the current training epoch is greater than the preset maximum epoch, the resultant trained model is subsequently extracted and utilized for testing purposes with the testing datasets. Alternatively, if the current training epoch is below the maximum threshold, the process returns to step (2) for continued training iterations.(4)The detection results of the final crack image testing datasets are output.

## 3. Experimental Verification

### 3.1. Dataset Preparation

This paper uses the Concrete Structure Spalling and Cracking (CSSC) dataset proposed by Li et al. [41] to rigorously substantiate the effectiveness and superiority of our GM-ResNet road crack detection model. Specifically, we use the concrete crack sub-dataset for model training, which comprises 522 real images of cracks annotated at the pixel level, and a total of 50,200 crack images are generated through data enhancement methods in this dataset, including images with and without cracks. The specific operation is to cut the real image of the crack into sub-parts of different sizes, such as 100 ∗ 100 and 130 ∗ 130, to generate a series of sub-images. In addition, we rotate the image and re-implement the entire process to generate more image information. By repeating the above operation, 50,200 sub-crack images can be generated from 522 real crack images. The training and testing datasets are divided randomly and non-overlap in an 8:2 ratio, with 40,160 crack images in the training dataset and 10,040 crack images in the testing datasets. Since the size of each image is inconsistent, we standardized the image dimensions to 224 × 224 pixels during training, accommodating variations in image size, to ensure uniformity in the model inputs. Considering the space constraints, we present ten select images as illustrative examples. Figure 8 serves as a representative visual, spotlighting an example image from the training crack dataset.

### 3.2. Model Training Parameter Settings

We know that the selection of optimal model parameters bears a significant influence on the ultimate training outcomes of the deep learning model. Therefore, after multiple experimental tests in this paper, we chose the stochastic gradient descent (SGD) with a momentum factor of 0.9 as the model optimizer and set the batch size to 64 and the weight attenuation to 0.001. The training regimen extends across 10 epochs, with the initial learning rate set to 0.01. In order to modulate the learning rate dynamically, the StepLR scheduler was introduced to decrease the learning rate to one tenth of the original every five epochs so as to avoid optimization bottlenecks in the training process that could lead to poor training accuracy. Table 1 shows the important parameters required for training the road crack detection model.

### 3.3. Model Training and Testing

After preparing the datasets and determining the parameters required for training the model, the training model can be started. The GM-ResNet is trained based on the pre-training model of the ImageNet large-scale image datasets. Training based on the weight of the pre-training model can speed up the training speed and make the training effect reach its optimum as soon as possible. As shown in Figure 9, the training accuracy and loss curve of the GM-ResNet road crack detection model are depicted. Notably, over a span of 10 training epochs encompassing 6280 iterations, the model’s training accuracy approaches a state of stability, yielding remarkable precision. This is accompanied by a downward trend in loss, reaching a minimal value. By observing the fluctuation state of the curve, we can see that the model has excellent stability and robustness in the training process. After 3000 iterations, the fluctuation of the accuracy rate and loss curve basically tends to be stable, with only occasional small oscillations. At the same time, it is often not comprehensive enough to assess the ability and performance of the holistic model by relying solely on the evaluation indicator of accuracy. Therefore, recall, precision and F1-score are introduced as auxiliary measurement indicators. The specific definitions and calculation rules of the three evaluation indicators are as follows:(a)True positive class (TP): the model correctly determines the input positive category samples as positive category samples.(b)True negative class (TN): the model correctly determines the input negative category samples as negative category samples.(c)False positive class (FP): the model mistakenly determines the input negative category samples as positive category samples.(d)False negative class (FN): the model mistakenly determines the input positive category samples as negative category samples.

Recall indicates the ratio of the number of true category samples correctly classified in the crack datasets to the whole quantity of all positive category samples. The recall formula is as follows:(7)$$Recall=\frac{TP}{TP+FN}$$

Precision represents the ratio of the true category samples correctly classified to the whole quantity of positive category samples. The precision formula is as follows:(8)$$Precision=\frac{TP}{TP+FP}$$

The F1-score is obtained by counting the harmonic average of the precision and recall, and its value range is [0, 1]. When the value of the F1-score is close to one, it indicates perfect precision and recall, and if the precision or recall is zero, the F1-score value is zero. The F1-score is calculated as follows:(9)$$F1-score=2\times \left(\frac{Recall\times Precision}{Recall+Precision}\right)$$

Therefore, the visualization curves of the three evaluation indicators of the GM-ResNet are shown in Figure 10. According to the figure below, when the GM-ResNet has achieved satisfactory effects in terms of recall, its precision and F1-score can be seen. Remarkably, as the training iterations advance, the F1-score steadily approaches unity, an encouraging indicator of the exceptional precision and recall achieved through the training process of the proposed model, which further verifies the robustness and supremacy of our model within the field of road crack detection.

In order to fully evaluate the superior performance of the proposed model, we present a comprehensive comparative analysis involving five distinct architectures: AlexNet, DenseNet, SqueezeNet, VGG16 and ResNet-34, as the experimental comparison models of the proposed GM-ResNet model. The training optimizer and learning rate scheduler of all the models are consistent with the previous text. Similarly, the model is trained using pre-training weights based on the ImageNet dataset to ensure fairness in the comparative experiment. The cross-entropy loss function is used as the training loss function for these five models. By inputting the crack image dataset into the models given above for training, we generate Figure 11, which portrays the training accuracy and loss histograms for the final epoch of all six models. Observing the following figure, we can see that the accuracy and loss of the AlexNet and DenseNet models after ten epochs of training are not ideal. In stark contrast, SqueezeNet exhibits the poorest performance, characterized by the lowest accuracy, the highest loss and the worst model performance. Meanwhile, VGG16 and ResNet-34 display promising results with an accuracy exceeding 97% and a corresponding minimal loss, and their model performance is good. Nonetheless, in contrast to alternative comparison methodologies, our proposed model consistently outperforms its counterparts in terms of training accuracy and loss. This observation reinforces the model’s superiority and underscores its capability to yield superior outcomes.

In the comparative analysis, recall, precision and F1-score serve as supplemental assessment metrics, corroborating the performance of the six models. In order to better evaluate the training performance of the models, the average values of these three evaluation indicators were taken on each epoch, and the results are shown in Table 2. Table 2 shows the specific average values of each epoch for the three evaluation indicators of the six models mentioned above. It can be seen from the table that the three evaluation indicators of the proposed GM-ResNet are superior to those of other comparative models, which further proves the effectiveness of the proposed model in the field of road crack detection.

In addition, since the values of TP, TN, FP and FN have already been calculated in previous work, it is natural to calculate the confusion matrix for model training through TP, TN, FP and FN and visualize the confusion matrix graph as another indicator for evaluating model performance. Serving as a synopsis of the classification prediction outcomes, the confusion matrix becomes instrumental in assessing the overall efficacy of the model training. Figure 12 presents the training confusion matrix diagrams for the six models. The deeper the background of the confusion matrix, the higher the classification accuracy of the model. It can be seen from the figure that, compared with the other comparison models, the proposed model achieves a high accuracy in the division of positive and negative samples, which further verifies the superiority of the proposed model in this paper.

Shadows in crack images are considered a special kind of interference, which can lead to some false detection during model detection. Shaded areas can be incorrectly classified as cracks, ultimately resulting in the degraded detection performance of the model and adversely affecting its detection accuracy. In order to verify the detection performance of the proposed model under shadow conditions, we add shadows to the dataset and compare the detection accuracy of the proposed model under shadow conditions and without shadow conditions compared with other models, as shown in Figure 13.

Looking at Figure 13, it can be seen adding shadows to the crack image does have a certain negative impact on the detection performance of the model. AlexNet, DenseNet, SqueezeNet, VGG16 and ResNet-34 all have significantly reduced detection accuracy. Although the detection accuracy of the proposed GM-ResNet crack detection model decreases, the amplitude is not high, and its accuracy is still the best compared with the comparison models, which further verifies the detection effect and robustness of the proposed model under shadow conditions.

In addition, in order to verify the impact of different modules of the GM-ResNet on the performance of the model, we supplemented the ablation experiment and conducted comparative tests without the GAM, without the MFCNN, without the GAM + MFCNN and the GM-ResNet, respectively. In order to enrich the test results, we also conducted the same test under the shadow condition, and the test results are shown in Figure 14.

By observing the ablation experimental results, it can be seen that the different module parts of the GM-ResNet proposed in this paper have a certain impact on the performance improvement of the model. Because it is under shadow conditions, several modules of the GM-ResNet play a great role in resisting shadow interference. The effectiveness and superiority of the proposed crack detection model are further verified through the ablation experiment.

Based on the analysis above, it can be seen that the GM-ResNet road crack detection model proposed in this paper has achieved commendable performance during the training process, and the comparative experiments with the other five models have further verified that this model has compelling superiority and viability. After the model training is completed, the test phase begins, and road crack classification and detection are carried out by inputting partitioned testing datasets into the pre-trained model. In a quest to validate the crack detection efficacy of our proposed model, the five models above were compared and classified for the same testing datasets. Figure 15 visually shows the crack detection and classification outcomes across the six models. By observing the figure below, we can see that the detection and classification effects of the five comparison models are not ideal. Notably, in the case of the intricate inaugural crack image, AlexNet, DenseNet, VGG16 and ResNet-34 exhibit a limited capacity to accurately pinpoint the segments of the crack area, and the efficacy of the crack detection is notably modest. VGG16 and ResNet-34 are better than AlexNet, DenseNet and SqueezeNet for simple single-crack images and can detect most cracks. Among them, the crack detection effect of SqueezeNet is the worst. Whether it is a complex crack or a simple crack, the model cannot detect and identify it correctly, indicating that the model has a serious sample representation problem and finds it difficult to correctly distinguish between positive and negative samples. By contrast, the GM-ResNet road crack detection model, as presented in this work, capitalizes on its ingenious architecture and judiciously chosen loss functions to proficiently discern complex and simple cracks. While its performance in detecting intricate cracks might not achieve perfection, its classification detection prowess remains unparalleled. This further substantiates the trustworthiness and adaptability of the model proposed in this paper.

After the training of the GM-ResNet crack detection model proposed in this paper, the image test set containing cracks is preprocessed and divided into smaller images of fixed area size, which are, respectively, input into the training model for prediction. If cracks are detected in the area, it will draw a colored rectangle in the original image to identify the detected cracks so as to determine the crack location. Based on the above work, we further calculate the crack length and angle information through Canny edge detection. For example, the specific crack information on the last three columns of the crack images in Figure 15 is shown in Table 3.

Although the relevant crack information of the crack image can be calculated, it is still a rough estimate of the crack information in the final analysis, and the model can not get very accurate, specific data. Therefore, in our future follow-up work, we hope to develop a crack detection model with higher efficiency and better effect and enable it to accurately calculate a series of related crack information from the crack images.

## 4. Conclusions

This paper proposes a road crack detection network framework based on a GM-ResNet, designed to significantly enhance detection accuracy. It is summarized as follows:(1)By introducing the GAM attention module to improve the feature extraction ability of the model, this mechanism enables comprehensive three-channel feature extraction and cross-dimensional interaction between the channels and spatial dimensions, avoiding a decrease in the feature extraction ability caused by the omission of useful feature information.(2)Relying on the deep network architecture, batch normalization layer and activation function layer in the MFCNN module to enhance the nonlinear fitting ability of the model and avoid overfitting, the generalization performance of the model is effectively enhanced.(3)The focal loss function is introduced to train the model to overcome the category imbalance problem of road crack detection; thus, the model can better distinguish between positive and negative samples and difficult and easy classification samples and improve the final detection performance of the model.(4)The experimental verification of the concrete crack image datasets also shows that the proposed model is significantly superior to other comparative models in terms of accuracy, recall and precision evaluation indicators, as well as in the output effect of crack detection, and has achieved significant progress, demonstrating the effectiveness and applicability of the model in the field of crack detection.

Although the GM-ResNet road crack detection framework proposed in this paper has achieved excellent performance, its detection performance for complex cracks is not yet perfect. Therefore, in future work, we will develop a more suitable detection model framework for the goal of complex crack detection in order to achieve precise complex crack detection results.

## Figures and Tables

**Figure 1 sensors-23-08369-f001:**
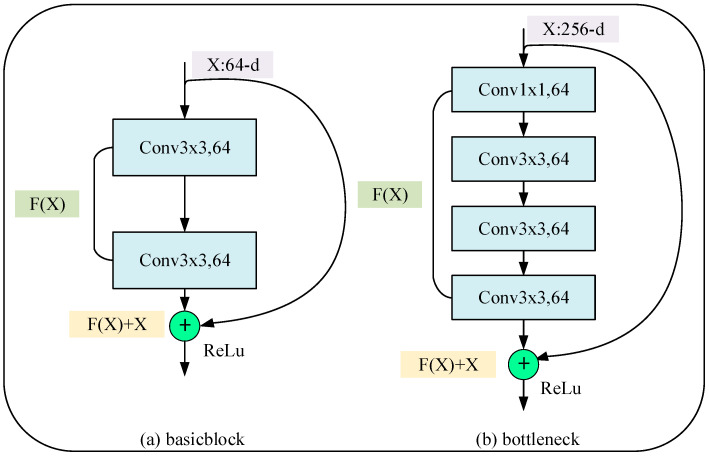
Residual block network structure diagram (**a**) for ResNet-18 and ResNet-34 and (**b**) for deeper networks, such as ResNet-50.

**Figure 2 sensors-23-08369-f002:**
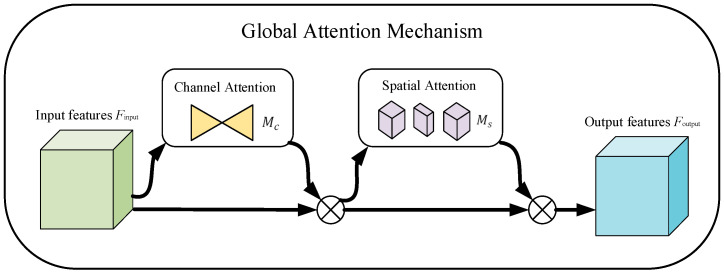
GAM network structure diagram.

**Figure 3 sensors-23-08369-f003:**
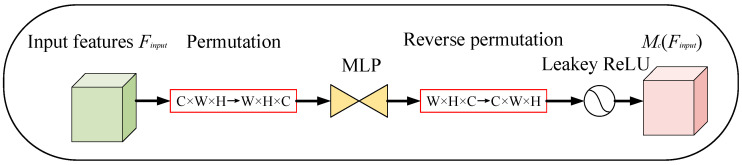
CAM network structure diagram.

**Figure 4 sensors-23-08369-f004:**
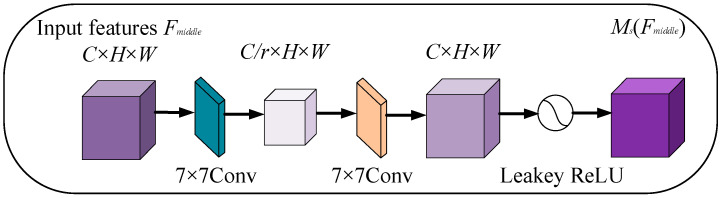
SAM network structure diagram.

**Figure 5 sensors-23-08369-f005:**
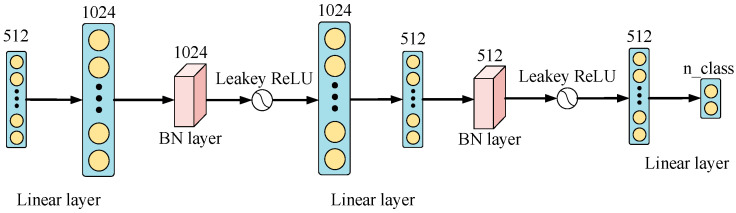
MFCNN network structure diagram.

**Figure 6 sensors-23-08369-f006:**
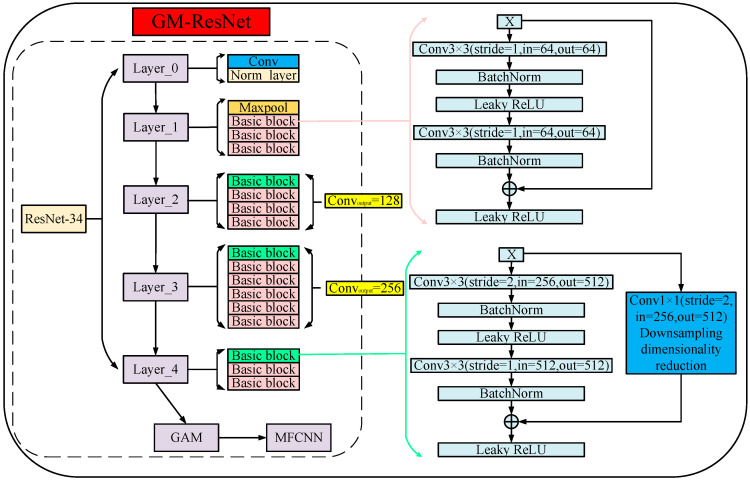
The GM-ResNet network structure diagram.

**Figure 7 sensors-23-08369-f007:**
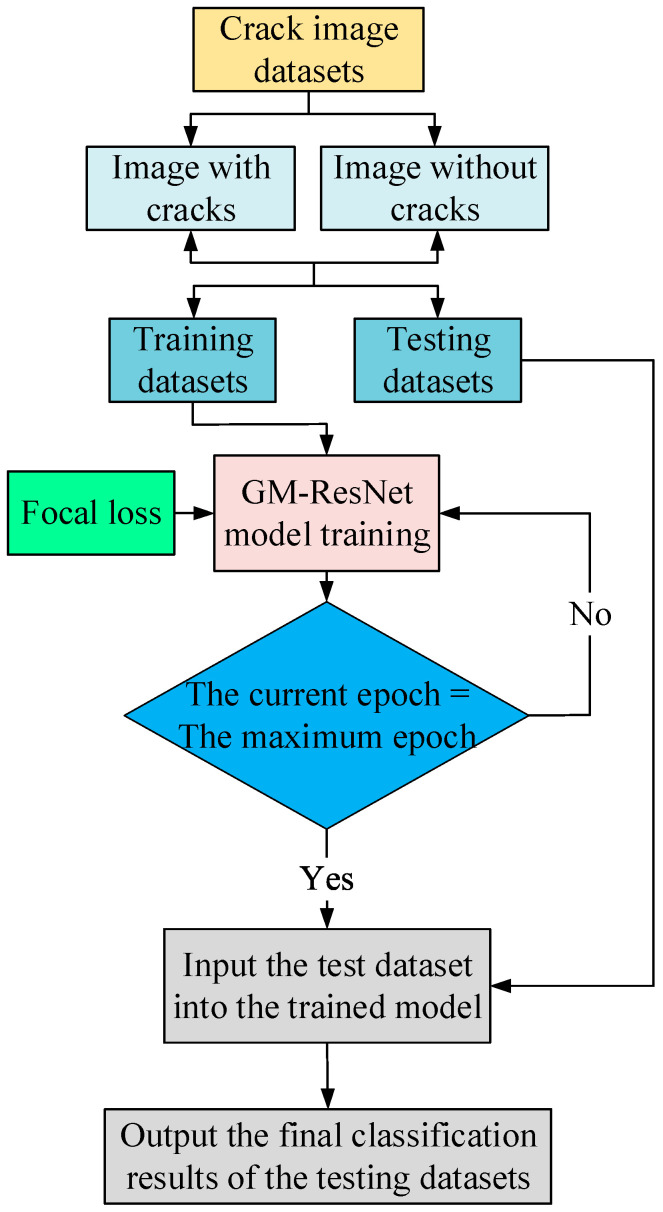
The flow chart of GM-ResNet road crack detection framework.

**Figure 8 sensors-23-08369-f008:**
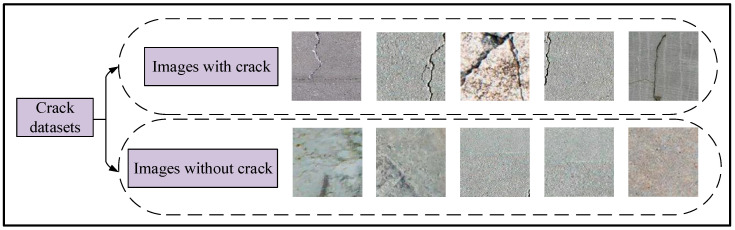
The example images of training crack datasets.

**Figure 9 sensors-23-08369-f009:**
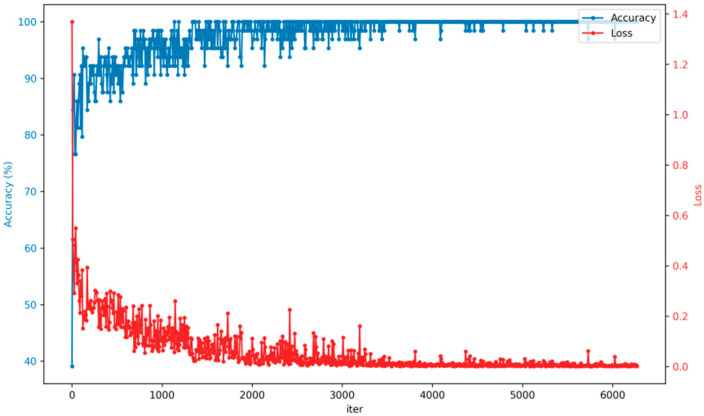
The training accuracy and loss curves of the GM-ResNet road crack detection framework.

**Figure 10 sensors-23-08369-f010:**
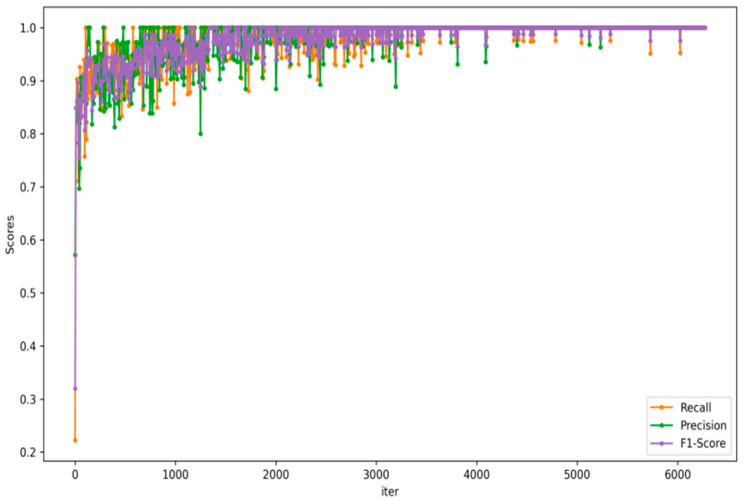
The training evaluation indicators curve of GM-ResNet road crack detection framework.

**Figure 11 sensors-23-08369-f011:**
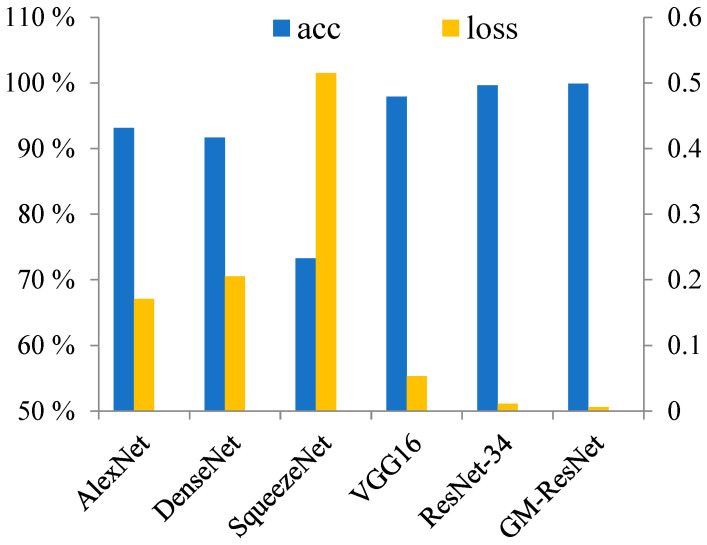
The accuracy and loss comparison histograms of six models.

**Figure 12 sensors-23-08369-f012:**
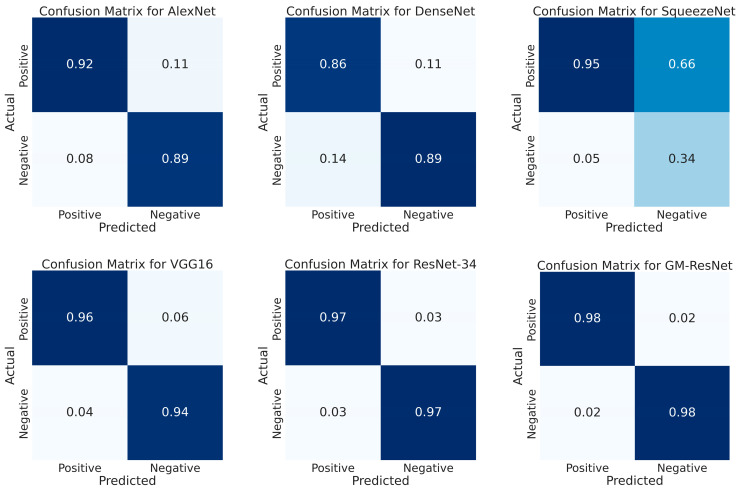
The training confusion matrix diagram of the six models.

**Figure 13 sensors-23-08369-f013:**
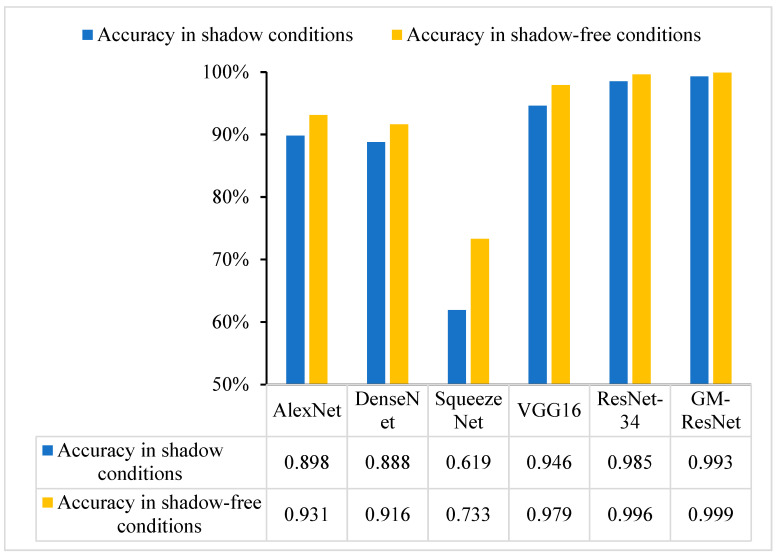
The histogram of ablation experiment comparison results.

**Figure 14 sensors-23-08369-f014:**
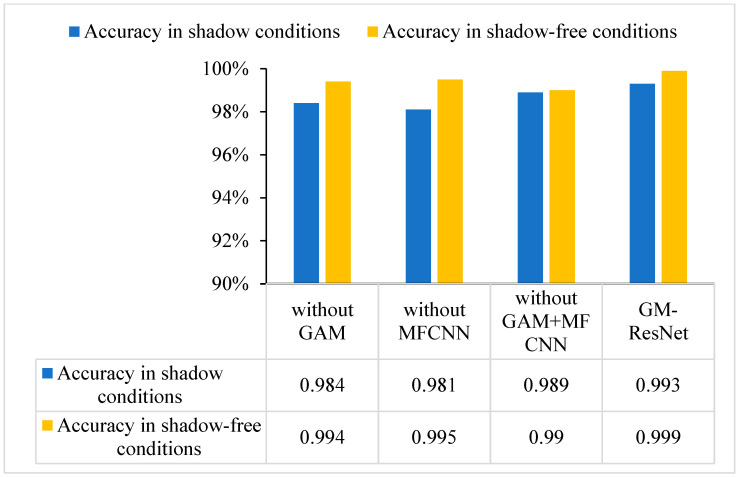
The histogram comparison of crack detection accuracy under shadow and shadow-free conditions.

**Figure 15 sensors-23-08369-f015:**
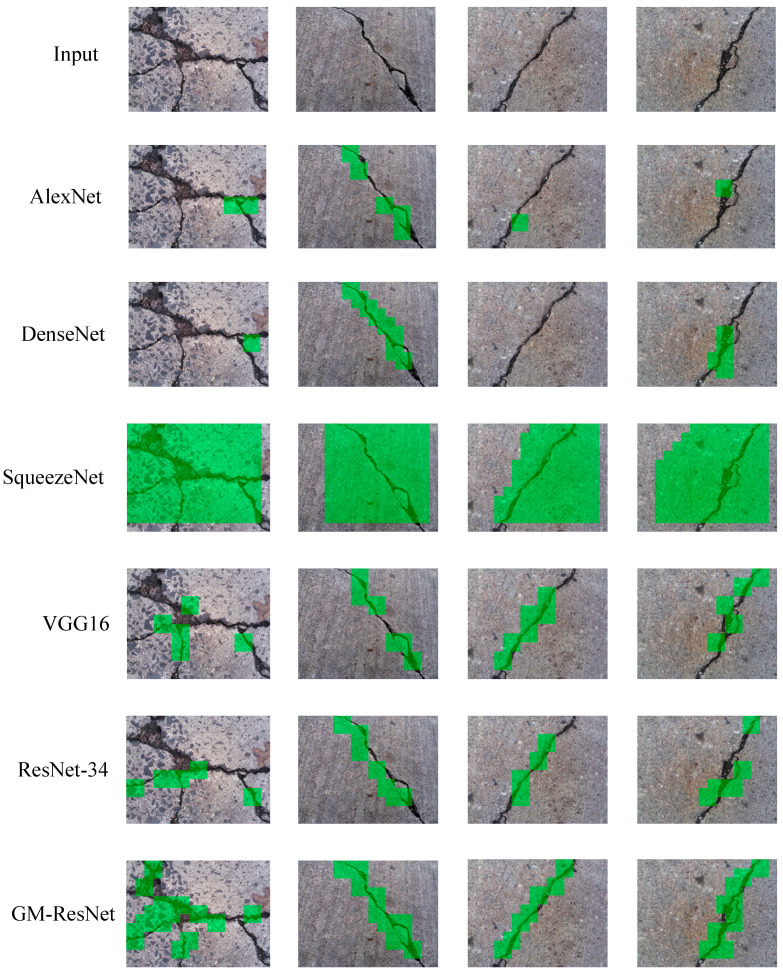
The crack detection effect diagram of six models.

**Table 1 sensors-23-08369-t001:** Important parameter settings of the model.

Parameter	Value	Parameter	Value
epoch	10	batch	628
batch size	64	optimizer	“SGD”
momentum type	“nesterov”	momentum	0.9
dampening	0.9	weight decay	0.001
lr scheduler	“StepLR”	learning rate	0.01
gamma	0.1	step size	5

**Table 2 sensors-23-08369-t002:** The average value of evaluation indicators of GM-ResNet road crack detection framework.

Model	Epoch	Recall	Precision	F1-Score	Model	Epoch	Recall	Precision	F1-Score
AlexNet	1	0.868	0.862	0.861	VGG16	1	0.913	0.883	0.895
	2	0.91	0.888	0.897		2	0.938	0.919	0.927
	3	0.915	0.9	0.906		3	0.947	0.934	0.939
	4	0.921	0.906	0.911		4	0.953	0.948	0.949
	5	0.926	0.912	0.917		5	0.959	0.954	0.955
	6	0.93	0.919	0.923		6	0.965	0.964	0.963
	7	0.932	0.921	0.925		7	0.97	0.97	0.969
	8	0.936	0.93	0.932		8	0.973	0.975	0.973
	9	0.937	0.932	0.933		9	0.979	0.98	0.979
	10	0.938	0.934	0.935		10	0.98	0.982	0.98
DenseNet	1	0.728	0.806	0.756	ResNet-34	1	0.899	0.89	0.892
	2	0.817	0.859	0.834		2	0.944	0.943	0.942
	3	0.843	0.88	0.859		3	0.963	0.966	0.964
	4	0.858	0.896	0.874		4	0.975	0.978	0.976
	5	0.872	0.905	0.886		5	0.985	0.986	0.985
	6	0.878	0.914	0.894		6	0.99	0.991	0.99
	7	0.886	0.921	0.901		7	0.992	0.992	0.992
	8	0.893	0.927	0.908		8	0.993	0.993	0.993
	9	0.901	0.934	0.916		9	0.995	0.995	0.995
	10	0.905	0.939	0.92		10	0.996	0.997	0.996
SqueezeNet	1	0.992	0.532	0.691	GM-ResNet	1	0.911	0.903	0.905
	2	0.986	0.557	0.709		2	0.958	0.951	0.953
	3	0.954	0.604	0.736		3	0.975	0.972	0.972
	4	0.948	0.63	0.753		4	0.985	0.985	0.985
	5	0.944	0.648	0.764		5	0.992	0.991	0.991
	6	0.942	0.661	0.773		6	0.996	0.996	0.996
	7	0.943	0.665	0.776		7	0.998	0.998	0.998
	8	0.942	0.679	0.785		8	0.999	0.998	0.998
	9	0.946	0.684	0.79		9	0.999	0.998	0.999
	10	0.946	0.684	0.791		10	0.999	0.999	0.999

**Table 3 sensors-23-08369-t003:** The specific value of crack information.

Image	Length (cm)	Angle (°)
Image (2)	84.601	105.289
Image (3)	85.624	75.195
Image (4)	86.327	73.440

## Data Availability

The data provided in this work are available from the corresponding author.

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
