# Peer review of "Intelligent Crack Detection Method Based on GM-ResNet"

_sensors, 2023, doi:10.3390/s23208369_

Round 1

Reviewer 1 Report

This manuscript presents an enhanced road crack detection network framework based on GM-ResNet, which matches the scope of the Journal. It contains some interesting results, which can be valuable to researchers in the industry and academia. There are however following issues to be addressed before the manuscript can be considered for publication on the journal:

1)      Some details should be provided on how the data enhancement methods are used to generate 50,200 crack images from 522 real images.

2)      Explain the meaning of the green colour in Fig.14.

3) Comment on the feasibility of the proposed method to provide more crack information such as the lengths and the angles of the cracks.

Author Response

Responds to the comments of Reviewer 1

Dear Reviewer and Editor,

We would like to sincerely thank you for taking the time to review our manuscript titled "Intelligent Crack Detection Method Based on GM-ResNet"(ID: sensors-2617975). Your valuable feedback and comments have been immensely helpful in improving the quality of our work. We have responded and explained one by one according to your comments, and marked the modified parts in yellow in the text, so that you can see our modified content more clearly. The main corrections in the paper and the responds to the reviewer’s comments are as flowing:

  1. Some details should be provided on how the data enhancement methods are used to generated 50200 crack images from 522 real images.

R: Thank you very much for your valuable comment. We have supplemented and added the details of the data generation method. This paper cuts the real image of cracks into sub parts of different sizes, such as 100 * 100 and 130 * 130, to generate a series of sub images. In addition, we rotate the image and re implement the entire process to generate more image information. By repeating the above operation, 50200 sub crack images can be generated from 522 real crack images.

  1. Explain the meaning of the green color in Fig.14.

R: Thank you very much for your valuable comment. The green boxes in Fig.14 refer to the model generating a predefined aspect ratio box when identifying cracks. The area covered by this box represents the portion where cracks exist, and the green color is used to highlight the location of the cracks. We can see from Fig. 14 that when a crack is found in a certain area of the image, a green box is used to indicate that there is a crack here. If we observe that there is no crack here, but there is a green box covering it, it indicates that the model has been detected incorrectly there.

  1. Comment on the feasibility of the proposed method to provide more crack information such as the lengths and the angles of the cracks.

R: Thank you very much for your valuable comment. We have added information on how to calculate crack length and angle information in the article. After the training of the GM-ResNet crack detection model proposed in this paper, the image test set containing cracks is preprocessed and divided into smaller images of fixed area size, which are respectively input into the training model for prediction. If cracks are detected in the area, it will draw a colored rectangle in the original image to identify the detected cracks, so as to determine the crack location. Based on the above work, we further calculate the crack length and angle information through Canny edge detection. For example, the specific crack information about the last three columns of crack images in Figure 15 is shown in the table 3.

Table 3. The specific value of crack information

Image

Length(cm)

Angle(°)

Image (2)

84.601

105.289

Image (3)

85.624

75.195

Image (4)

86.327

73.440

Although the relevant crack information of the crack image can be calculated, it is still a rough estimate of the crack information in the final analysis, and cannot get very accurate specific data. Therefore, in the future follow-up work, we hope to develop a crack detection model with higher efficiency and better effect, and enable it to accurately calculate a series of related crack information of the crack image.

The above is all of our responses and modifications to the comments of reviewer. We once again thank you and the reviewers for your professional opinions and valuable suggestions. We are deeply impressed by the positive experience of this review process. We have made modifications to the paper according to your suggestions and hope that these modifications can meet the requirements of your journal. We have submitted the revised paper and response letter as attachments to you. If you need any additional information or documents, or have further suggestions, please let us know. Thank you again for your professional review. We look forward to working with you to successfully publish this work.

Yours sincerely,

Prof. Xiangyang Xu

21 Sep., 2023

School of Rail Transportation, Soochow University.

Reviewer 2 Report

The paper presents an interesting subject. The following aspects must be improved:

-the related work section must contain existing methods with obtained results  

-section 2.1 and 2.2 are better to be eliminated - they are presenting some general aspects about CNN and ResNet that are not relevant for the article

-the ablation study must be made by using existing methods for crack detection, methods that are described in related work section

-is it possible to use the method in real time? what is the inference time?

-how shadows can influence the detection of cracks

-what do you mean by complex cracks that are not correctly detected?

Author Response

Responds to the comments of Reviewer 2

Dear Reviewer and Editor,

We would like to sincerely thank you for taking the time to review our manuscript titled "Intelligent Crack Detection Method Based on GM-ResNet"(ID: sensors-2617975). Your valuable feedback and comments have been immensely helpful in improving the quality of our work. We have responded and explained one by one according to your comments, and marked the modified parts in yellow in the text, so that you can see our modified content more clearly. The main corrections in the paper and the responds to the reviewer’s comments are as flowing:

  1. The related work section must contain existing methods with obtained results.

R: Thank you very much for your valuable comment. Because the GM-ResNet crack detection model proposed in this paper is optimized and improved on the basis of ResNet, our original intention is to make a brief introduction to CNN and ResNet, the backbone networks of the model proposed in this paper, in the relevant basic work part of this article, so as to lay the foundation for the subsequent improvement of this paper. The existing methods and results have been described in detail in the introduction section.

  1. 1 and 2.2 are better to be eliminated-they are presenting some general aspects about CNN and ResNet that are not relevant for the article.

R: Thank you very much for your valuable comment. We sincerely accept your suggestions on this issue. We deleted the contents of sections 2.1 and 2.2, and added some relative contents into the chapter 3, in order to ensure the integrity of the content structure of the paper. The modified parts are as follows:

  1. The proposed GM-ResNet road crack detection framework

This section initiates with the concepts of the GAM and the proposition of a MFCNN as a replacement for the ResNet fully-connected layer. Subsequently, the focal loss function is introduced. Finally, amalgamating GAM and MFCNN within the ResNet-34 framework, we present a novel GM-ResNet architecture tailored specifically for the task of road crack detection.

2.1. ResNet

The rapid evolution of deep learning technology has propelled the widespread utilization of CNN as an exemplar network model. Its pervasive influence spans across a spectrum of domains, including facial recognition[25], intelligent transportation systems[26] and cancer classification[27], or in the fields of structural health monitoring[28] and fault diagnosis[29]. The ubiquity of CNN in these realms substantiates their pivotal standing as an architectural cornerstone within the deep learning domain. Emerging as an unparalleled and triumphant paradigm, CNNs have assumed the mantle of the most extensively adopted and efficacious deep learning network model. The breakthrough by AlexNet[30] in 2012, marked by extraordinary performance on expansive image datasets, ignited a transformative wave in the trajectory of deep learning. This pioneering success catalyzed the formulation of a suite of innovative and exceptional variants of CNN, such as GoogleNet[31], VGGNet[32], DenseNet[33] and SqueezeNet[34], each making indelible marks across diverse tasks, emerged as emblematic exemplars, showcasing remarkable accomplishments in distinct arenas. Many experts and scholars are committed to improving the feature representation ability by continuously stacking deeper network architectures to achieve a better performance of the model. However, as the depth of the model increased, a perplexing observation emerged: contrary to expectations, model performance exhibited diminishment. The reason for this was that gradient vanishing and explosion within the intricate network of hidden layers and the troubling issue of network degradation collectively contributed to the pronounced deterioration of model efficacy. The proposal of ResNet[35] successfully solved this challenge. The ResNet, notably diverging from conventional CNN, adds the output of all the layers before the active function to the output of the current layer through the skip connection structure of the residual blocks. This strategic fusion empowers the network to transmit the gradient directly to the previous network through the skip connection during backpropagation, effectively mitigating the issue of network degradation. Thereby, avoiding the problems of gradient explosion and disappearance during the training process of the model, which resulted in poor model performance.

Assuming that  denotes the original input data of the model, then  is the output after passing through the residual block. The calculation formula of the identity map  is as follows:

By using the equation above, an identity map can be constructed to convert the function  required for the training and fitting of the original model into . This simple addition can significantly enhance the model training speed and efficacy, and precludes network degradation due to excessive layering. The residual block encompasses two distinct types: the basic block and the bottleneck. Figure 2 is the network structure diagram of the residual block.

Figure 1. Residual block network structure diagram. (a) For ResNet-18 and ResNet-34, (b) for deeper networks, such as ResNet-50.

  • GAM
  • MFCNN
  • Focal loss function
  • GM-ResNet Network Framework for Road Crack Detection
  1. The ablation study must be made by using existing methods for crack detection, methods that are described in related work section.

R: Thank you very much for your valuable comment. The GM-ResNet proposed in this paper is essentially a crack detection model based on CNN, so we have carried out a detailed performance comparison test between the proposed model and a series of other CNN based models in the experimental verification part. The following part is the ablation experiment content that we supplement this paper.  In addition, in order to verify the impact of different modules of GM-ResNet on the performance of the model, we supplemented the ablation experiment, and conducted comparative tests on without GAM, without MFCNN, without GAM+MFCNN and GM-ResNet respectively. In order to enrich the test results, we also conducted the same test under the shadow condition, and the test results are shown in the figure.

Figure 14. The histogram comparison of crack detection accuracy under shadow and shadow-free conditions

By observing the ablation experimental results, it can be seen that the different module parts of GM-ResNet proposed in this paper have a certain impact on the performance improvement of the model. Because it is under shadow conditions, several modules of GM-ResNet play a great role in resisting shadow interference. The effectiveness and superiority of the proposed crack detection model are further verified by the ablation experiment.

  1. Is it possible to use the method in real time? What is the inference time?

R: Thank you very much for your valuable comment. The crack detection model proposed in this paper is not a real-time detection method. It relies on offline processing of crack datasets to extract crack image features and differentiate them from background features for the purpose of crack detection. Our future work will focus on further developing a new method that can efficiently and real-time detect cracks.

  1. How shadows can influence the detection of cracks?

R: Thank you very much for your valuable comment. We have supplemented the experimental section under shadow conditions, as follows: Shadows in crack images are considered a special kind of interference, which can lead to some false detection during model detection. Shaded areas can be incorrectly classified as cracks, ultimately resulting in degraded detection performance of the model and adversely affecting detection accuracy. In order to verify the detection performance of the proposed model under shadow conditions, we add shadows to the dataset and compare the detection accuracy of the proposed model under shadow conditions and without shadow conditions compared with other models, as shown in the figure.

Figure 15. The histogram of ablation experiment comparison results

Looking at the above figure, it can be seen that after adding shadows to the crack image, it does have a certain negative impact on the detection performance of the model. AlexNet, DenseNet, SqueezeNet, VGG16 and ResNet-34 all have significantly reduced detection accuracy. Although the detection accuracy of the proposed GM-ResNet crack detection model decreases, the amplitude is not high, and its accuracy is still the best compared with the comparison model, which further verifies the detection effect and robustness of the proposed model under shadow conditions.

  1. What do you mean by complex cracks that are not correctly detected?

R: Thank you very much for your valuable comment. The complex cracks that I referred to as not being correctly detected refer to crack images in the first column of Figure 14 that are not just a single crack, but rather a more complex composition of several different cracks. Therefore, complex crack images impose higher requirements on the model's detection performance. In this paper, both the proposed crack detection method and its comparative methods fail to detect all of these complex cracks. Hence, they are referred to as not being correctly detected. In this study, comparatively speaking, the proposed method still achieves the best detection results. Thanks a lot.

The above is all of our responses and modifications to the comments. We once again thank you and the reviewers for your professional opinions and valuable suggestions. We are deeply impressed by the positive experience of this review process. We have made modifications to the paper according to your suggestions and hope that these modifications can meet the requirements of your journal. We have submitted the revised paper and response letter as attachments to you. If you need any additional information or documents, or have further suggestions, please let us know. Thank you again for your professional review. We look forward to working with you to successfully publish this work.

Yours sincerely,

Prof. Xiangyang Xu

21 Sep., 2023

School of Rail Transportation, Soochow University.

Round 2

Reviewer 2 Report

Since all my comments were addressed, I recommend to publish the paper.